

# A new workflow for the automated measurement of shape descriptors of rocks

Eszter Fehér[1,2], Balázs Havasi-Tóth[1,3], and Balázs Ludmány[4]

[1]MTA-BME Morphodynamics Research Group
[2]Department of Mechanics, Materials and Structures
[3]Department of Fluid Mechanics
[4]Department of Control Engineering and Information Technology

**Correspondence:** Eszter Fehér (fehereszter@szt.bme.hu)

**Abstract.** Shape properties of rocks carry important geological information about their origin, and they may also provide a window to study the abrasion processes forming their geometry. The number of mechanical equilibria is a significant property with a profound mathematical background that could reveal the secrets hidden in the artifacts of Nature. Although it is easy to count by hand, the automation of its measurement is not a straightforward task. A new workflow is introduced for the fast and

efficient measurement of geometrical properties, including the number and location of stable and unstable equilibrium points of rocks based on a portable 3D scanner combined with computer software that can analyze the resulting point cloud. The technique allows for the fast examination of statistically sufficient sample sizes without the need for transportation or storage of the specimens. A previously hand-measured set of pebbles and fragments was used as a reference for the validation of the method, and its effectiveness is demonstrated through the examination of beach pebbles carried out in Kawakawa Bay, New

Zealand.

## 1 Introduction

Measuring shape characteristics of sedimentary particles is an effective way to formulate hypotheses about their history based on the mathematical models of abrasion processes. Theoretical results show that analysis of the shape of rocks may answer

questions related to their place of origin, travel distance, and it can also reveal details of the abrasive forces that contributed to the final geometry. The most general shape descriptors are the axis ratios ($c/a$, $b/a$ for $a > b > c$) (Zingg, 1935) and the roundness or isoperimetric ratio ($I$) that expresses how close a shape is to a sphere. Traditional measurement techniques often incorporate personal factors or rely on the verbal characterization of the shape (Wentworth, 1923; Boggs, 2001) to approximate these values. The number of stable ($S$) and unstable ($U$) mechanical equilibria of a particle as a shape descriptor gained

significant attention recently (Domokos et al., 2009; Miller et al., 2014; Domokos et al., 2015; Novák-Szabó et al., 2018) and



it is not only insensitive for small measurement errors, but it also has a rich mathematical literature (Grayson, 1987; Domokos et al., 2015; Domokos and Lángi, 2019).

It is beneficial to switch from the traditional hand-measurements to automated shape analysis of the particles to avoid personal bias. Recently, several works appeared aiming to reduce subjectivity by automatically calculating shape properties from 2D digital images of the particles (Roussillon et al., 2009; Durian et al., 2007; Cassel et al., 2018; Cheng et al., 2018), 3D laser scanning (Hayakawa and Oguchi, 2005; Anochie-Boateng et al., 2013) or X-ray CT (Deiros Quintanilla et al., 2019). Development in sensors and 3D cameras induced a technological revolution in shape detection in many fields from the poultry industry (Chan et al., 2018) to ballast in railway track structures (Anochie-Boateng et al., 2013). 3D scanning was successfully applied in geology as well for the analysis of rocks. Among the obvious advantages of these techniques, most of the papers reported some of their drawbacks compared to hand-measurements. The workflow usually consists of collecting the specimens and transporting them to the laboratory to carry out the scanning. There are numerous options for 3D scanning, and most of them require a well-prepared laboratory environment. Professional scanners are usually limited in portability, rather expensive, and developed for scanning one object at a time, aiming high precision. As a result, the scanning process is relatively slow, and taking the postprocessing into account, it can take up weeks to capture a statistically significant amount of specimens. As a result, the scanning process is relatively slow, and taking the postprocessing into account, it can take up weeks to capture a statistically significant amount of specimens. There exist techniques aiming at portability that does not require expensive devices. Such a method is the 3D image reconstruction from multiple photos taken from different directions of the object, but it requires more complex postprocessing; thus its is even more time-consuming. Consequently, there is a trade-off between portability and the amount of work needed to obtain the final image. Finally, these scanners are usually developed for industrial applications, and many of them are struggling to capture small objects.

The purpose of our work is twofold. Firstly, we introduce a new workflow of 3D scanning, that solves many of the issues regarding the 3D analysis of rocks and allows for the fast and efficient scanning of large sets of specimens. Secondly, we combine the technique with an existing algorithm that can calculate not only the traditional shape parameters, but it can also analyze a point cloud captured from a three-dimensional object and determine its equilibrium points computationally. Our method is based on a portable scanner Structure Sensor Mark I and software built on a recently published algorithm (Ludmány and Domokos, 2018) to determine the equilibrium points of a 3D point cloud. We utilize the fact that the algorithm works with the convex envelope of the point cloud and that the polygonal approximation of the object might result in the creation of artificial equilibrium points that can be neglected. As a result, aiming extremely high precision during the scanning is unnecessary, and simple portable scanners can also be satisfactory for the task. Portable scanners were successfully used in the industry for shape detection (Chan et al., 2018). The main advantage of our technique is that there is no need to transport the specimens and store them for further analysis in the laboratory, hundreds of specimens can be captured in a few days near their origin, and the resulting 3D images can be analyzed later by the software. Such a technique could advocate data sharing and collaboration between experimental and theoretical groups.

The structure of our paper is the following: in Sect. 2, we introduce the scanning process and the remarkably short post-processing resulting in the 3D point cloud, and the software carrying out the shape analysis is described in detail. Section 3 is



devoted to the validation of the technique on a reference set of specimens that were previously analyzed by hand-measurements. In Sect. 4, we illustrate the robustness of the method in a field-measurements at the Kawakawa Bay, New Zealand. Finally, we summarize our results in Sect. 5.

## 2   New method

### 2.1   3D scanning


The scanning of three-dimensional objects requires the visibility of the entire surface of the object. In case of former geometry reconstruction techniques, a series of photos or partial scans need to be performed of the pebbles placed on flat surfaces from different angles before the assembly of the final geometry. Unfortunately, the merging procedure is not straightforward and usually has to be done manually, which is often extremely time-consuming due to the removal of the undesired surfaces attached to the pebble geometry. Moreover, it is often a problem that most of the handheld scanners suffer from issues when recording small objects and easily lose the track of pebbles under a certain size. Our aim was to elaborate a scanning technique, which allows the uninterrupted scanning of pebbles resulting in a seamless geometry without the need of any manual postprocessing. One of the most crucial aspect of such a scanning technique is how the pebble should be fixed in space without even a small parcel of its surface being covered by the holder. To overcome this problem, we constructed a holder of two $0.25 \times 0.25$ m frames shown in Fig. 1 with tight webs. As long as the strings forming the webs are thin enough to be invisible for the scanner device, the frames manage to hold the pebble stationary leaving plenty of space for the observer to scan each sides.



In order to avoid the scanner to be rotated around the frame, we designed the assembly so that it can be rotated smoothly on a tripod. After fixing the pebble in the frame, the scanning can be performed in one continuous session using the scanner Structure Sensor Mark I. Since the scanner is quite compact, one can hold it by hand during the entire scanning. However, focusing on reproducibility, the results shown in the present work have been obtained using a camera crane fixed on another tripod. The complete layout of the two tripods with the rotating frame and the crane is presented in Fig. 1.


It is crucial for the scanner to never lose the track of the pebble during the scanning process, hence the crane is designed to move the scanner in such a way that the pebble always remains in the center of the view regardless of the crane angle. As Fig. 2 shows, the scanner defines a scanning domain in which it generates the geometry based on the distance field, while everything outside the domain is completely ignored. After the scanning process is initiated, it is essential for the successful scanning to never let the domain out of the sight of the scanner. Since the scanner does not have built-in gyroscope or accelerometer, it computes its relative spatial position and orientation merely based on the recorded three-dimensional point cloud. Therefore, in order to facilitate a robust and accurate orientation, we placed four spherical foam balls over the frames providing sufficiently large reference surfaces for the orientation regardless of the pebble size. Lacking of global reference points, the domain is automatically attached to the rotating frames. The scanning starts at the upper position of the crane, and after a full 360-degree rotation of the specimen, the crane is moved to the bottom position, which is followed by a second 360-degree turn of the specimen. This method ensures that every side of the specimen is captured. The scanner is connected to a computer running Skanect Pro recording the geometry.





Earth **Surface**
**Dynamics**
Discussions



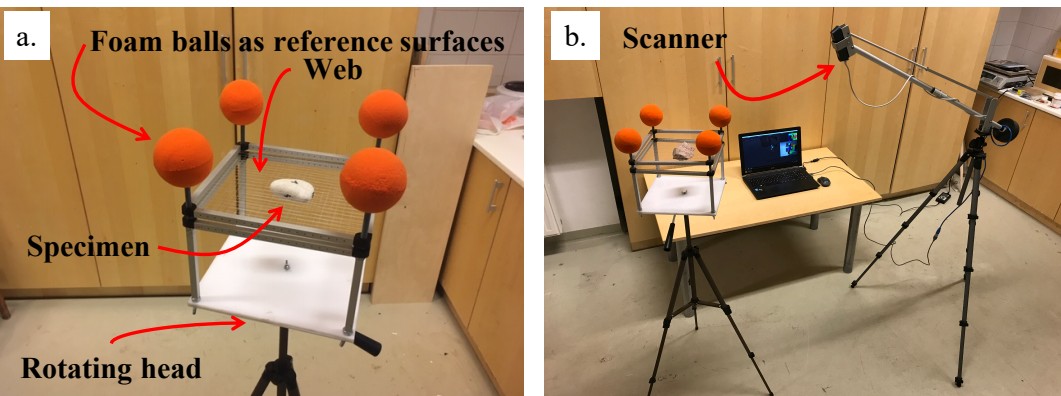

**Figure 1.** Photograph of the rotating head equipped with the web and the reference points (*a.*), and the complete layout with the two tripods (*b.*).

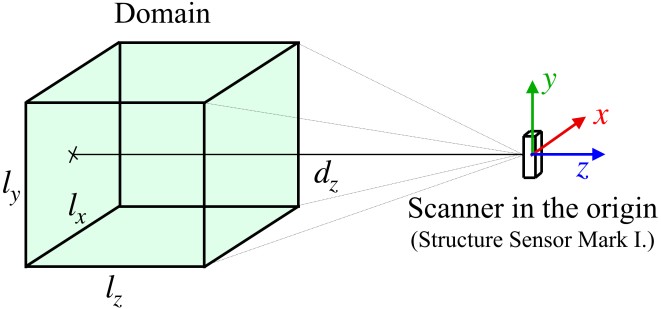

**Figure 2.** The position and sizes of the scanning domain; $d_z = 0.8$ m, $l_x = l_y = l_z = 0.5$ m.

Finally, the geometry containing the specimen and its surroundings with the frame of the web and the reference spheres are exported to stereolithography format. Since the web remains invisible, the pebble geometry is independent of its surroundings, the pebble can be easily extracted using a CAD software (e.g. Blender). Automation of this step is a possible development in the future. The preview and the final geometry are shown in Fig. 3.

With this procedure, one pebble can be scanned under two minutes, and the postprocessing takes up less than a minute. The new scanning technique is compared to the existing methods in Table 1. Although the proposed method defeats the existing methods in portability, speed, and price, the quality of the final point cloud is expected to be lower compared with a fixed laser scanner. However, we expect exponential progress on the development of 3D scanners.

## 2.2 Automated shape analysis

The scanning process results in individual .stl geometries of each specimen. The software, based on the algorithm of Ludmány and Domokos (2018), can handle input in Object File Format (OFF) or STereoLitography (STL) format. It is available in Lud-



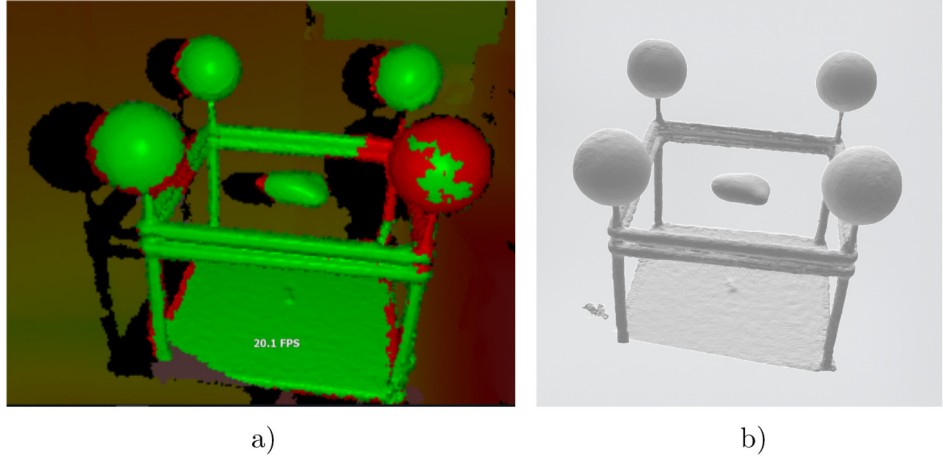

a)                 b)

**Figure 3.** Scanning of the object and its environment. a) Outlook of the scene during the scanning process. b) The resulting object.

| technique | portability | recording speed | postprocessing speed | price | references |
|---|---|---|---|---|---|
| fixed 3D laser/optical scanning | - | slow (30 min) | slow (30 min) | expensive | Hayakawa and Oguchi (2005); Anochie-Boateng et al. (2013) |
| 3D from 2D photos | x | slow (40-200 images) | slow (1 hour) | cheap | Tomasi and Kanade (1992) |
| CT, X-ray | - | slow (7 particle/day) | fast | very expensive | Deiros Quintanilla et al. (2019) |
| proposed method | x | fast (2 min) | fast (1 min) | cheap | |

**Table 1.** Comparison of the existing techniques and the proposed method in terms of portability, recording and postprocessing speed, and price.

mány (2020b). It was showed in Domokos et al. (2011b, a), that artificial equilibrium points can appear due to the polyhedral approximation of the surface, and the equilibrium points of the physical object correspond to the convex hull of the point cloud. As a result, the surfaces with faces contained in the input files are ignored, and a triangulated convex hull of the points is constructed right away.

    The algorithm calculates level sets of the convex hull and represents the body by $M$ contour lines enclosing points lying at

least a distance $s_i$ from an arbitrary internal reference point, which is the centroid by default (Fig. 4). The equilibrium points are closed contour lines containing no other contour lines having a larger area than $\rho$ percent of the total surface area. This restriction provides a smoothing of the surface to avoid the effects of the polyhedral approximation of the smooth body. Both





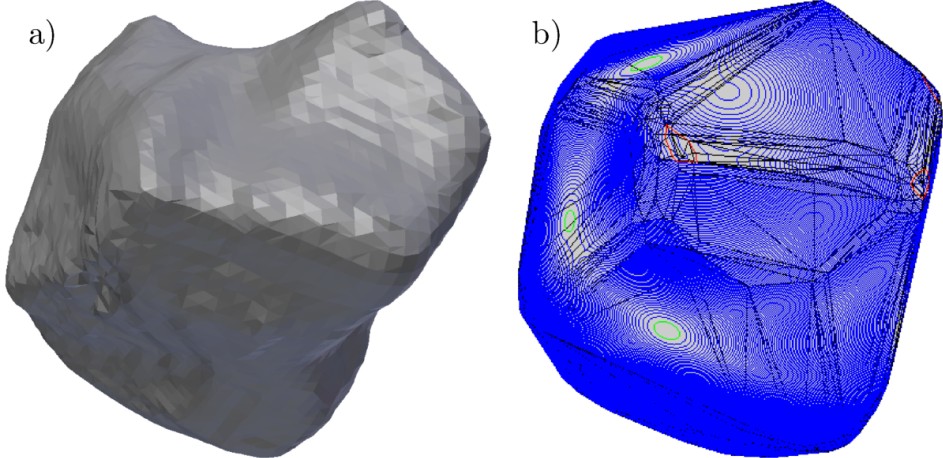

**Figure 4.** a) The extracted geometry of the object. b) Visualization of the contour lines of the convex hull and the equilibrium points of the scanned geometry.

$M$ and $\rho$ are input parameters of the algorithm. The center of mass of the real object can differ from the centroid of its convex hull for concave shapes or inhomogeneous objects. To consider the uncertainties, the software is able to consider multiple $m$

number of reference points lying inside a sphere of radius $R$ around the centroid and calculate their minima, maxima and average $S_{avg}$ and $U_{avg}$. Apart from the number of equilibrium points, the algorithm calculates multiple geometrical properties of the convex hull. The main outputs are the following:

- the number of stable ($S$) and unstable ($U$) equilibrium points

- surface area ($A$) and volume ($V$) of the convex hull

- longest ($a$), second-longest ($b$) and third longest axis ($c$) and their ratios ($c/a$, $b/a$)

- circumference ($C_p$) and area of the planar projection with the largest possible area ($A_p$)

- isoperimetric ratio ($I$) of the convex hull in relation to the bounding sphere and the ellipsoid with diameters $a$, $b$ and $c$

- isoperimetric ratio of the planar projection with the largest possible area ($I_p$) in relation to the bounding circle and the ellipse with diameters $a$ and $b$

There are two modes of operation: the user can either open a single file or a batch of files. In the first case, the convex hull is displayed in a 3D viewer, and the equilibrium points found by the algorithm are visualized. This allows for the fast and straightforward comparison of the scanned geometry to the real object. The second mode is a bulk evaluation of multiple files using predefined values of the input parameters $M, \rho, m, R$ in a batch file (in CSV format). In case of $m = R = 0$ the reference point is the centroid; otherwise, the aggregation method is also required. The results can be saved in CSV format, and the





statistical analysis can be carried out in Excel or Matlab. The computation is parallelized, and depending on the parameters and the number of pebbles, it takes a few minutes.

The computer program presented here is merely a user-friendly graphical interface on top of the algorithm, which is separated in a function library available in Ludmány (2020a). It does not require the point cloud to be in a `.off` or a `.stl` format. Consequently, this technology can be easily integrated into any other application.

## 3 Benchmark

We fitted the input parameters of the software to hand-measured data of 367 pebbles and fragments from various locations, including different rock types and colors. From the output parameters of the algorithm listed in Sect. 2.2, we restricted the benchmark test on $a, b, c, S, U$, which can be measured reliably. The hand-measurements were carried out under laboratory conditions by one person for consistency. We scanned the specimens with the method described in Sect. 2.1 and evaluated the geometries with the algorithm using different parameter combinations of $M, \rho, m, R$. Firstly the $a, b, c$ values were used to check the scanning quality since they are independent of the input parameters and cannot be improved. From $S$ and $U$, the latter is less straightforward to measure and cannot be treated with equal importance. As a result, we fitted the input parameters to the measured number of stable equilibrium points and expected higher difference for $U$. We defined two error norms:

$$\bar{e}_S = \frac{1}{N} \sum_{i=1}^{N} \frac{|S_i^m - S_i^c|}{S_i^m}, \tag{1}$$

$$\bar{e}_U = \frac{1}{N} \sum_{i=1}^{N} \frac{|U_i^m - U_i^c|}{U_i^m}, \tag{2}$$

$$e_{\bar{S}} = \frac{|\sum_{i=1}^{N} S_i^m - \sum_{i=1}^{N} S_i^c|}{\sum_{i=1}^{N} S_i^m}, \tag{3}$$

$$e_{\bar{U}} = \frac{|\sum_{i=1}^{N} U_i^m - \sum_{i=1}^{N} U_i^c|}{\sum_{i=1}^{N} U_i^m}, \tag{4}$$

where $e_S, e_U$ are the average of the errors specimen by specimen and $e_{\bar{S}}, e_{\bar{U}}$ are the errors of the average for a set of $N$ pebbles. For statistical analysis, the average error is more relevant.

6 different sets of pebbles (A1, C1, K1, N1, TA1, T2) and 1 set of fragments (F) were analyzed and evaluated using the same parameter combinations. By comparing $a, b, c$, the limits of the scanner were also tested: specimens that were too small resulted in significant differences in $a, b, c$; as a result, we limited the volume in $V = 1000\ mm^3$. The size of the specimens ranged from $1000 - 21000\ mm^3$. The best fit of the parameters was $M = 200, \rho = 0.05, m = 10, R = 0.001$, which resulted in good agreement between the measurements and computations for pebbles. Figure 5 shows the error of the averages for each sets and 2 summarizes the data of each set. As we can see in Fig. 5, the error is relatively high in the case of fragments, which can be attributed to the precision of the scanner, which is 0.5 mm in the best possible case according to the manufacturer. The





scanner did not capture the edges of the fragments appropriately, leading to a loss of equilibrium points. Nevertheless, portable scanners are continuously developed and we expect better agreement in the future.

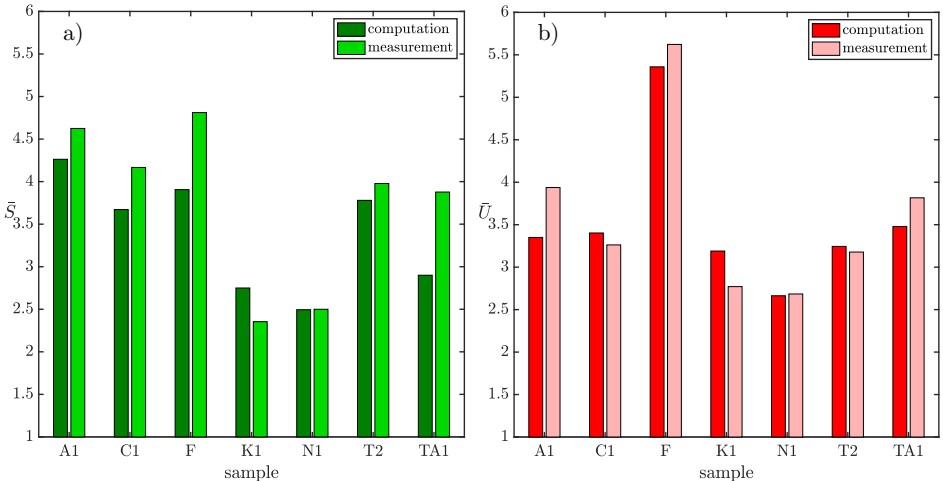

**Figure 5.** Average of the computed and measured number of stable and unstable equilibrium points for seven sets of samples using the best parameter combinations. a) Comparison of $\bar{S}$. b) Comparison of $\bar{U}$.

| sample | $\bar{S}^c$ | $\bar{S}^m$ | $\bar{U}^c$ | $\bar{U}^m$ | $\bar{e}_S$ [%] | $\bar{e}_U$ [%] | $e_{\bar{S}}$ [%] | $e_{\bar{U}}$ [%] | $N$ |
|--------|-------------|-------------|-------------|-------------|-----------------|-----------------|-------------------|-------------------|-----|
| A1 | 4.26 | 4.63 | 3.35 | 3.94 | 22.98 | 27.51 | 7.84 | 12.7 | 16 |
| C1 | 3.67 | 4.17 | 3.4 | 3.26 | 22.99 | 15.01 | 11.89 | 3.37 | 42 |
| F | 3.91 | 4.81 | 5.36 | 5.62 | 24.03 | 21.47 | 18.82 | 5.49 | 53 |
| K1 | 2.75 | 2.35 | 3.19 | 2.77 | 26.75 | 27.08 | 16.81 | 17.79 | 48 |
| N1 | 2.49 | 2.5 | 2.66 | 2.68 | 13.1 | 13.28 | 0.21 | 0.84 | 76 |
| T2 | 3.78 | 3.98 | 3.24 | 3.18 | 24.84 | 26.37 | 4.97 | 1.68 | 45 |
| TA1 | 2.9 | 3.88 | 3.48 | 3.82 | 25.2 | 16.34 | 25.21 | 8.74 | 49 |

**Table 2.** The computed and measured S and U values and the errors in percentage. $N$ is the number of specimens in the set.

Note that $S$ and $U$ are round numbers so that the error can be relatively high. Due to the averaging evaluation, we expect

that the method won't detect extremities, e.g., it is not able to detect monostatic bodies. However, due to the uncertainty in the exact location of the centroid, the consideration of multiple reference points is inevitable.





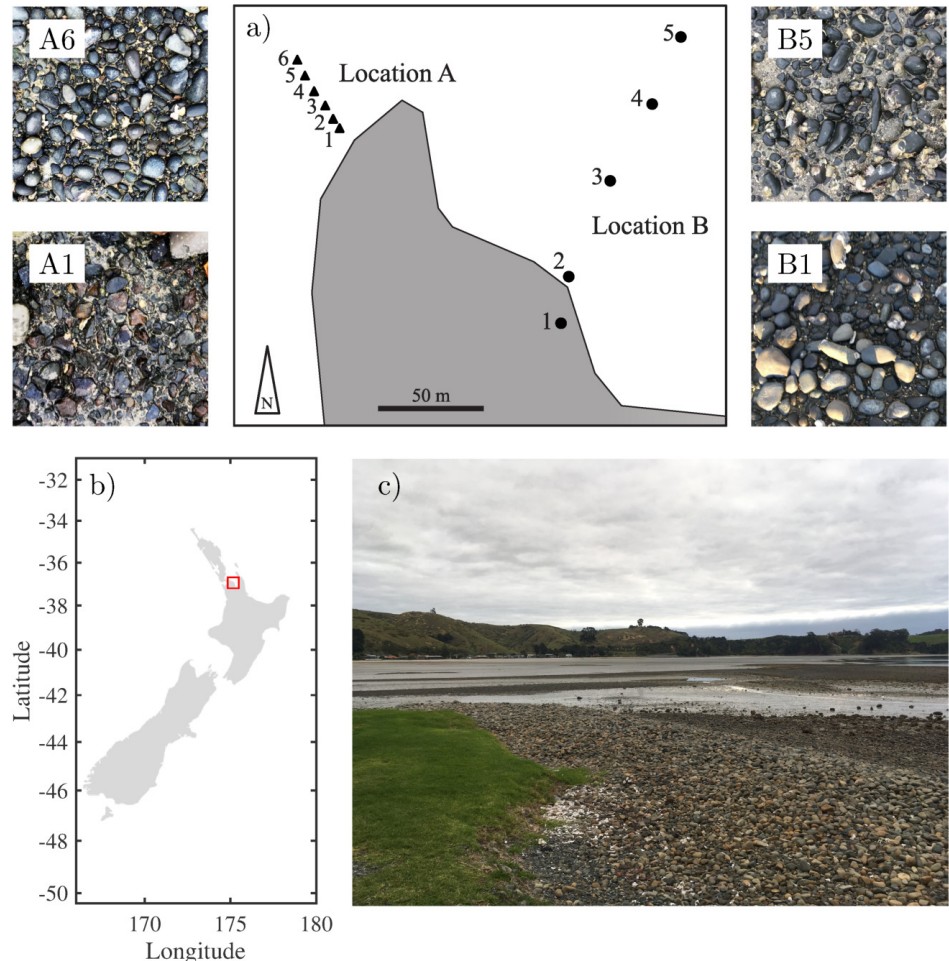

**Figure 6.** Kawakawa Bay, New Zealand. a) Location A and B. b) Map of New Zealand with the Kawakawa Bay highlighted by a red square. c) A photograph of the Kawakawa Bay.

## 4 Field study

We demonstrate the robustness of the technique on beach pebbles from Kawakawa Bay, New Zealand 6. We performed the technique of Wolman (1954) for collecting 20-40 randomly chosen pebbles at each location. Firstly, we chose two locations and

appointed measurement points lying at equal distances along a line. A measurement point represented a line segment parallel to the coastline. The collecting person was moving from side to side, and the collecting process stopped when the satisfactory amount of pebbles was obtained. We scanned the collected samples with the proposed method and evaluated the geometries using the input parameters determined in Sect. 3.

Location A consisted of six measurement points lying approximately 8 meters far from each other in the direction perpen-

dicular to the coastline. The pebbles were slightly embedded in the sand; therefore, we expected small transportation. We found

Earth **Surface**
**Dynamics**
Discussions

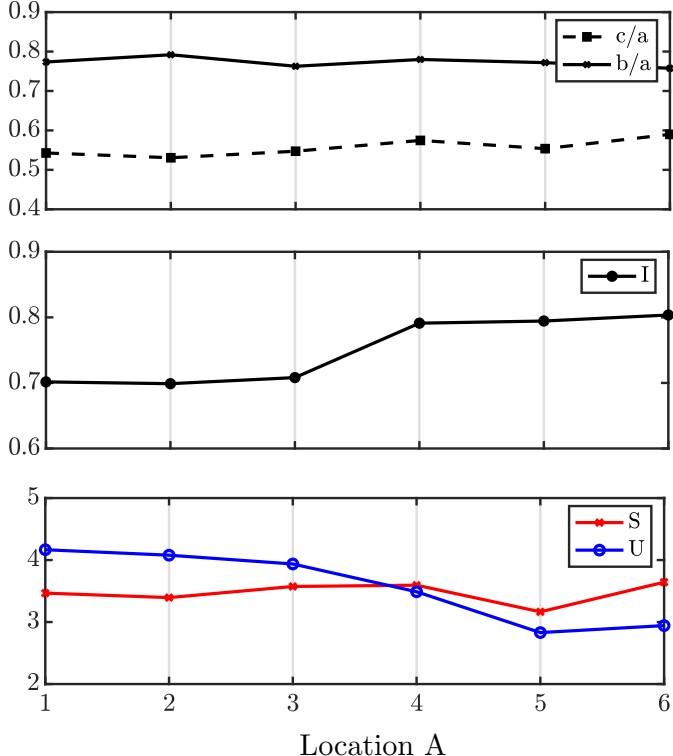

**Figure 7.** Shape properties evaluated on the scanned geometries of the samples collected from six measurement points of Location A. Location A1 is the closest to the coast. The results show that from location A1 to A6 the particles gradually change their shape. The number of unstable equilibrium points decreases, the isoperimetric ratio increases, and the other parameters stay approximately constant.

fragments at location A1 near the coast and rounded pebbles at location A6, but there was no visible difference between them in terms of size distribution. We assumed that the particles along Location A have the same origin, but those that are lying farther from the coast have small sharp edges and corners eroded. Otherwise, we expected no significant difference between rocks along the line. The analysis supported our assumptions. The aspect ratios $c/a, b/a$ and $S$ stayed approximately constant

along the line, but U decreased, and I increased. The number of specimens was 20-30 at each location; the size distribution was between $11500 - 108000 \, mm^3$.

   Location B consisted of five points lying from approximately 38 m far from each other. The path was also perpendicular to the coastline. Comparing location B1 and B5, we expected more complex abrasion processes. At location B5, the number of elongated pebbles seemed to be significantly more significant compared to location B1 based on visual inspection. The

scanning analysis proved this assumption. Both the aspect ratios decreased, meaning that two sizes of the pebbles decreased more than the longest size, leading to an elongated geometry. Subsequently, $I$ also decreased with $S$ and $U$ decreasing. The number of specimens was 15-30 and the size distribution was between $5000 - 170000 \, mm^3$.





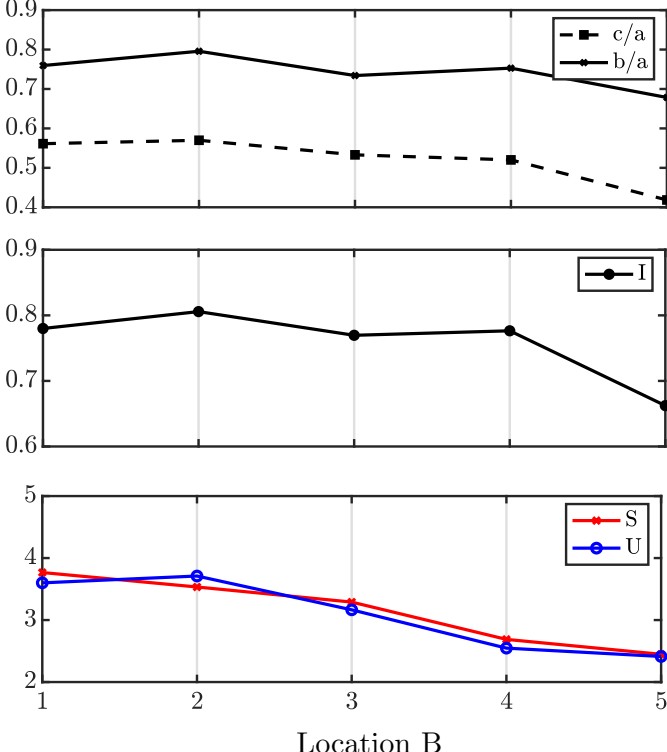

**Figure 8.** Shape properties evaluated on the scanned geometries of the samples collected from five measurement points of location B. The decrease in all of the measured shape properties show that moving farther from the coast, the particles become elongated.

Although the benchmark test in Sect. 3 showed, that using the Structure Sensor Mark I sharp edges cannot be recorded and results in a slightly decreased $S$ and $U$, using this technique, we managed to catch the gradual changes at both locations. We collected and scanned approximately 400 pebbles and fragments under three days without the need for transportation of the pebbles to the laboratory. Moreover, the dataset is available for further analysis.

## 5   Conclusions

A new scanning technique was introduced that can record and evaluate hundreds of specimens under a short period. By introducing portable scanners to geological applications, it becomes unnecessary to transport the collected samples to the laboratory. Moreover, the postprocessing takes only a few minutes using the suggested setup, and it can be easily automated. Shape parameters such as the axis ratios, the isoperimetric ratio and the number of static equilibria are evaluated using from the convex hull of the point cloud that represents the surface with contour lines. The technique is proved to be an excellent alternative to hand-measurements. We expect accelerated development on the hardware side, Structure Sensor Mark II. is already available, and the compatible Skanect Pro version is on the go, that will extend the range of scannable pebble sizes and could provide



better precision for fragments. It is also expected, that this measurement technique would encourage collaborations between international groups through easy data sharing and storing. As the mathematical background of abrasion models and shape descriptors develops, there might appear new perspectives that can be examined on existing pebble geometries.

A possible extension of the method could be the scanning of multiple pebbles at the same time. If the scanner is connected to an iPad, then it is able to record the texture of the surface, e.g., it can record the ID written on the specimen. This way, the
pebbles can be easily distinguished from each other and extracted in Blender. Structure Sensor Mark II comes with a wide lens camera that can record the texture without an iPad connection. Depending on the size of the pebble, it would be possible to scan 4-5 pebbles at the same time, which could significantly reduce the overall scanning time.

*Code availability.* The open source libraries of the algorithm used in this work can be downloaded from Ludmány (2020a) and Ludmány (2020b).

*Video supplement.* A video demonstration of the new method is available at Fehér et al. (2020).

## 1 GPS coordinates of the measurement points of the field study

**Table 1.** GPS coordinates of the measurement points at location A.

| ID | latitude | longitude |
|----|----------|-----------|
| A1 | 36°56'49.1"S | 175°09'51.0"E |
| A2 | 36°56'49.0"S | 175°09'50.9"E |
| A3 | 36°56'48.7"S | 175°09'50.7"E |
| A4 | 36°56'48.5"S | 175°09'50.5"E |
| A5 | 36°56'48.3"S | 175°09'50.3"E |
| A6 | 36°56'48.1"S | 175°09'50.2"E |

*Author contributions.* The steps of the 3D scanning process were developed by authors Balázs Havasi-Tóth and Eszter Fehér, and Balázs Ludmány developed the automated shape analysis algorithm. The benchmark tests and the field study was carried out by Balázs Havasi-Tóth and Eszter Fehér. All authors contributed to writing of the document and interpreting the results.

*Competing interests.* The authors declare no competing financial interests.



**Table 2.** GPS coordinates of the measurement points at location B.

| ID | latitude | longitude |
| --- | --- | --- |
| B1 | 36°56'52.1"S | 175°09'55.3"E |
| B2 | 36°56'51.4"S | 175°09'55.4"E |
| B3 | 36°56'49.9"S | 175°09'56.2"E |
| B4 | 36°56'48.7"S | 175°09'57.0"E |
| B5 | 36°56'47.7"S | 175°09'57.6"E |

*Acknowledgements.* The authors would like to thank Krisztián Halmos for carrying out the hand measurements and for helping with the 3D scanning; Gábor Domokos for the fruitful discussions and his help in the field study; Bernd Krauskopf for helping to find a location to test the method; Géza Tóth for helping to build the camera crane. This research was supported by the NKFIH Hungarian Research Fund Grant 134199 and of Grant BME FIKP-VÍZ by EMMI is kindly acknowledged.



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
