# Peer review of "A new workflow for the automated measurement of shape descriptors of rocks"

_Earth Surface Dynamics, 2020_

## Referee Comment (RC1) · Anonymous Referee #1 · 8 Jun 2020

I thank the authors for their work on the manuscript entitled "A new workflow for the automated measurement of shape descriptors of rocks". This manuscript develops a new method 1) to scan individual pebbles in the field using 3D scanners and 2) to describe theses pebbles using automatic shape analysis. This approach is novel and could possibly lead to some future interesting results on the description of pebbles and their evolution in the environment.

However, I believe this manuscript, in its current form, needs to be thoroughly revised to be considered for publication. Indeed, 1) the description of the approach lacks some important aspects, 2) the statistical analysis in the results section is not convincing enough and 3) the field study seems rather unsubstantiated. I detailed these points below (general comments) and then I give some more specific comments. I also feel

that a proper discussion section is lacking to discuss the limitation of the approach, how it can positively influence current researches, how it compares with other grain analysis methods and to describe the implication of the findings of this study. Despite this, I strongly believe that with some significant work, this manuscript can become an important contribution to the field of grain shape analysis. So I encourage the authors to see this review as an opportunity rather than a negative outcome.

General comments:

1) Automatic shape analysis - This part is too focused on the technical aspect of shape analysis (which software, which files, how to use the software), and lacks a clear and well-structured description of what are equilibrium points (including stable and unstable ones) and how these equilibrium points are identified. For instance, the only definition given of an equilibrium point is (lines 105-106) "The equilibrium points are closed contour lines containing no other contour lines having a larger area than p percent of the total surface area." This a rather technical and very brief description of what are equilibrium points, which still let me wonder what you mean by equilibrium points. I agree I could go and read papers on equilibrium points, but I doubt that other readers will do this effort, moreover this a central concept for this manuscript. I would be please if the author can 1) explain what is an equilibrium point (in a non-technical manner) and 2) link this theoretical or physical description to a technical one and 3) explain how the proposed algorithm identifies equilibrium points. A figure describing the different steps of the used algorithm would probably be helpful.

2) Benchmark – Statistical comparison between the results obtained with hand measurements and automatic shape analysis is only performed using average value of metrics for each pebble sets (average of the number of stable points, etc) or average errors. Because average values can hide some information, I would also like to see the maximum and minimum differences and the standard deviation. I would also like to see a figure simply plotting $S_m$ as a function of $S_c$ (same for U, a, b, c) for all of the 367 pebbles using a specific color for each class of pebbles. If everything goes well,

there should be little spread compared to the 1:1 line.

3) Field study – Are the changes observed on Figure 7 and 8 statistically significant? There is no uncertainty analysis that can support the idea that the slight changes observed depart from the random hypothesis. This is a major limitation, as all the interpretations made in this section relies on the pre-supposed robustness of these changes. Moreover, I would also like to see a description of how grain size (D50 or D90) changes in between the different sites. Overall, I find this part quite unsubstantiated with many field inferences that appear quite unjustified or not demonstrated.

Specific comments:

All Figures - Figure captions are way too rudimentary and incomplete. For instance, on Fig. 4, what is the color scale on panel a? What are the red, blue, green and black colors on panel b?

Abstract - The abstract introduces well the work done in this paper, but, in my opinion, is not sufficiently focused on the main results.

Abstract - line 2 - Please rephrase this sentence which is too undefined for an abstract

Page 1 lines 14-16 - References are lacking to support this statement

Page 1 lines 19-20 - It would be worth giving a brief description of what is the definition of stable and unstable mechanical equilibria of a particle, as many readers of Esurf might not be familiar with these concepts.

Page 2 – lines 32-33 - I don't necessarily agree with this statement. Portative terrestrial Lidars now allow to scan very quickly many "specimens" with a relatively low amount of post-processing compared to some photogrammetric approaches.

Page 2 – lines 37 – "3D image reconstruction" - If you mean surface topographical reconstruction from 3D images, please say it. 3D image reconstruction can be interpreted in quite different manners

Page 2 – lines 37-38 – "Such a method is the 3D image reconstruction from multiple photos taken from different directions of the object, but it requires more complex post-processing; thus its is even more time-consuming." SFM can be relatively quick and post-processing can be automated, so I am not convinced this statement is correct.

Page 3 – lines 81-82 - It might be worth mentioning that new scanners (such as Mark II) have an IMU and that this limitation might not be relevant anymore.

Page 4 – line 95 - I don't agree on the portability. This setup seems less (or not more) portable than a terrestrial lidar that does not necessarily require to have a complex reference frame neither. The only advantage that I see is the scanning of the whole pebble.

Page 4 – line 96 – This statement is rather unsubstantiated

Page 6 – line 108 - Could you briefly state here what is the effect of changing M or rho ?

Page 6 – line 111 - Are we supposed to know what these two variables, Uavg and Savg, are?

Page 6 – line 124 - Which aggregation method are you referring to? Why is aggregation required? This is unclear.

Page 7 – line 125 – "Excel or Matlab" - I am not sure it is required to cite any software here, especially if they are referred to for simple table analysis.

Page 7 – lines 131-132 - I suggest sharing these 3D data on a server to make them open

Page 7 – lines 136-137 – "From S and U, the latter is less straightforward to measure and cannot be treated with equal importance. As" This sentence is unclear – please rephrase

Page 7 – lines 137-138 – "As a result, we fitted the input parameters to the measured

number of stable equilibrium points and expected higher difference for U." Should the optimization also consider the distance to the true location of the stable and unstable points? Detecting the correct number of stable points is indeed required, but if the location of these points does not match the real one, this is still problematic.

Equations 1-4, Figure 5 and Table 2 – see general comment 2

Page 9 – lines 148-150 - "We performed the technique of Wolman (1954) for collecting 20-40 randomly chosen pebbles at each location." 20 to 40 pebbles is a rather small number to infer any robust metrics on grain shape or size.

Page 9 – line 165 - "The pebbles were slightly embedded in the sand; therefore, we expected small transportation." Why? I don't get the argument here.

Page 10 – lines 167-168 – "We assumed that the particles along Location A have the same origin, but those that are lying farther from the coast have small sharp edges and corners eroded." This lacks a proper justification.

Page 10 – line 173 – "Comparing location B1 and B5, we expected more complex abrasion processes." Why?

Figure 6 and 7 – See general comment 3

---

## Referee Comment (RC2) · Anonymous Referee #2 · 13 Jul 2020

After carefully read the entitle manuscript "Review for A new workflow for the automated measurement of shape descriptors of rocks" it appears clear that the content is not much more than just having placed a Structure sensor on a rotating table. The Structure sensor being a commercial product delivered with a processing software, I do not see any novelty or any technical of scientific advance. As for the result analysis there is not much else than basic statistics on geometric parameters. In consequences, I do not see any real scientific content that meets eSurf criteria and I think the paper should be rejected in its current status. Notwithstanding, I suggest to the authors to redesign their aims and their methods as having automatic extraction of grain/pebble shape might be useful as mentioned in the introduction.

---

## Editor Comment (EC1) · Eric Lajeunesse (Editor) · 13 Jul 2020

Dear Authors,

I have now received two anonymous reviews of your manuscript. Both reviews raise major issues which make it hard to recommend revisions at this stage. Based on these reviews, I recommend you not to proceed with the present submission, but to prepare a thoroughly revised manuscript, that you might resubmit to Esurf in the future.

Sincerely yours,

Eric Lajeunesse
* * *
[Figure]

2020.

---

## Author Comment (AC1) · 10 Aug 2020

Dear Reviewer,

Thank you for the encouragement regarding our work and the detailed suggestions. We have decided not to proceed with the present submission and to prepare a revised manuscript for resubmission. We are going to address your comments and suggestions as well.

---

## Author Comment (AC2) · 10 Aug 2020

Dear Reviewer,

Thank you for your time and feedback. We have decided to prepare a revised manuscript where we are going to address your comments and suggestions as well.

---

## Author Comment (AC3) · 10 Aug 2020

Dear Editor,

Thank you for your time and feedback. We have decided not to proceed with the present submission and to prepare a revised manuscript for resubmission.
* * *